# Medical, Health and Wellness Tourism Research—A Review of the Literature (1970–2020) and Research Agenda

**DOI:** 10.3390/ijerph182010875

**Published:** 2021-10-16

**Authors:** Lina Zhong, Baolin Deng, Alastair M. Morrison, J. Andres Coca-Stefaniak, Liyu Yang

**Affiliations:** 1Institute for Big Data Research in Tourism, School of Tourism Sciences, Beijing International Studies University, Chaoyang District, Beijing 100024, China; zhonglina@bisu.edu.cn (L.Z.); dengbaolin0225@163.com (B.D.); liyu__yang@163.com (L.Y.); 2Greenwich Business School, Old Royal Naval College, University of Greenwich, London SE10 9SL, UK; a.coca-stefaniak@greenwich.ac.uk

**Keywords:** medical-health-wellness tourism, bibliometric analysis, thematic analysis, research agenda

## Abstract

Medical, health and wellness tourism and travel represent a dynamic and rapidly growing multi-disciplinary economic activity and field of knowledge. This research responds to earlier calls to integrate research on travel medicine and tourism. It critically reviews the literature published on these topics over a 50-year period (1970 to 2020) using CiteSpace software. Some 802 articles were gathered and analyzed from major databases including the Web of Science and Scopus. Markets (demand and behavior), destinations (development and promotion), and development environments (policies and impacts) emerged as the main three research themes in medical-health-wellness tourism. Medical-health-wellness tourism will integrate with other care sectors and become more embedded in policy-making related to sustainable development, especially with regards to quality of life initiatives. A future research agenda for medical-health-tourism is discussed.

## 1. Introduction

In 1841, Thomas Cook organized a tour of 570 people to travel from Leicester to Loughborough’s hot springs [1]. This was the first historically documented tour arranged by a travel agent. However, far earlier, people in Ancient Greece used to travel considerable distances for medical treatment [2]. Thus, the pursuit of health and medical care has been an essential reason for travel for centuries.

Today, people continue to travel in the pursuit of relaxation, for health reasons, as well as fitness and well-being [3]. As a response to this growing demand, countries, medical providers, and hospitality and tourism organizations are adapting to offer a broader set of medical, health, and wellness tourism experiences.

The concept of medical-health-wellness tourism has emerged relatively recently as a scholarly field of enquiry in tourism [4,5,6]. Although it has been pointed out that travel medicine has existed for 25 years [7], much of the research related to this has traditionally focused on medical aspects with inadequate consideration given to travel or tourism. Medical-health-wellness tourism can be classified into two primary categories according to a tourist’s choice - obligatory or elective. Obligatory travel occurs when required treatments are unavailable or illegal in the place of origin of the traveler and, as a result of this, it becomes necessary to travel elsewhere to access these services. Elective travel is usually scheduled when the time and costs are most suitable, and the treatments may even be available in the travelers’ home regions [8]. Other studies have classified these forms of travel and tourism into specific types based on the purpose of the treatment, such as dental tourism [9], stem cell tourism [10], spa tourism [11], springs tourism [12], IVF treatment [13], hip and knee replacements, ophthalmologic procedures, cosmetic surgery [5], cardiac care, and organ transplants [14].

A consensus is yet to be established on the definitions and contents of medical-health-wellness tourism, and how they interact, including their potential overlaps. Medical travel and tourism, health tourism, wellness tourism, and other similar terms (e.g., birth tourism, cosmetic surgery tourism, dental tourism) tend to be investigated separately in tourism research [15,16,17,18,19,20]. Notwithstanding the apparently disconnected nature of published research in this field, medical-health-wellness tourism has become much more popular for a variety of economic, cultural, lifestyle and leisure reasons [11,21,22]. Given their rapid development, it seems appropriate to conduct a comprehensive review of the definitions, history, typologies, driving factors, and future directions for these forms of tourism.

This study firstly reviews existing scholarly research through a meta-analysis of medical-health-wellness publications in the context of tourism (Section 2). Then, the method used to analyze the data collected from ISI Web of Science is outlined in Section 3, followed by a discussion of the research findings (Section 4). Finally, in Section 5, the conclusions, future research directions, and limitations of the study are presented.

## 2. Scholarly Reviews and Meta-Analyses of Medical, Health and Wellness Tourism

Previous reviews of the literature and meta-analyses have contributed to clarifying the overall understanding of medical-health-wellness tourism. Existing literature reviews tend to be very broad, spanning health-oriented tourism, medical tourism, sport and fitness tourism, adventure tourism, well-being (Yang sheng in Chinese) tourism, cosmetic surgery tourism, spa tourism, and more.

Medical tourism is an expanding global phenomenon [15,23,24]. Driven by high healthcare costs, long patient waiting lists, or a lack of access to new therapies in some countries, many medical tourists (mainly from the United States, Canada, and Western Europe) often seek access to care in Asia, Central and Southern Europe, and Latin America [25,26,27]. There are potential biosecurity and nosocomial risks associated with international medical tourism [28]. One research study collected 133 electronic copies of Australian television programs (66 items) and newspapers (65) about medical care overseas from 2005 to 2011 [29]. By analyzing these stories, the researchers discovered that Australian media coverage of medical tourism was focused geographically mainly on Asia, featuring cosmetic surgery procedures and therapies generally not available in Australia. However, people tend to engage with medical tourism for a broad range of reasons. In some cases, it is better service quality or lower treatment costs that prevail. In other cases, treatments may not be available locally, or there are long patient waiting lists for non-emergency medical care. Some 100 selected articles were reviewed and categorized into different types of medical tourism depending on the medical treatments they involved, such as dentistry, cosmetic surgery, or fertility work [25]. An analysis was done on 252 articles on medical tourism posted on the websites of the Korean Tourism Organization and the Korean International Medical Association [30]. This work enhanced the understanding of medical tourism in Korea as well as identifying the key developmental characteristics. Another research study detailed patient experiences in medical travel, including decision making, motivations, risks, and first-hand accounts [31]. A literature review was conducted on international travel for cosmetic surgery tourism [5] and it concluded that the medical travel literature suffered from a lack of focus on the non-surgery-related morbidity of these tourists.

Another set of authors defined health tourism as a branch of tourism in general in which people aim to receive specific treatments or seek an enhancement to their mental, physical, or spiritual well-being [32]. This systematic literature review assessed the value of destinations’ natural resources and related activities for health tourism. It was argued that most of the research on health tourism has focused on travel from developed to developing countries, and that there is a need to study travel between developed nations [33].

Wellness tourism is a key area of relevant research as well [34]. One research study reviewed trends in wellness tourism research and concluded that tourism marketing had so far failed to tap into the deeper meaning of wellness as a concept [35]. The emergence of health and wellness tourism was explored with their associated social, political, and economic influences [13]. A review was conducted of the development of wellness tourism using the concept of holistic wellness tourism where it was found that the positive impacts of this type of tourism on social and economic well-being were key to its rising levels of popularity [36]. 

All in all, although earlier literature reviews provide invaluable insights into medical-health-wellness tourism, there is a lack of studies that approach this concept in a holistic way. This research seeks to redress this balance by delivering a holistic review of the literature with the following objectives in mind: (1) investigating international journal articles across the typologies of tourism outlined above; (2) identifying influential scholars that have significantly contributed to this field; and (3) summarizing key trends in markets, industry development and promotion, as well as policy-making and impacts. In order to achieve this, a systematic review was conducted to analyze research articles in medical-health-wellness tourism published over a 50-year period from 1970 to 2020.

## 3. Methods

### 3.1. Data Collection

A two-step approach was adopted for the development of a database of publications for analysis with CiteSpace. The first step involved a search for relevant, high-quality refereed articles in medical-health-wellness tourism. Several academic journal databases, within tourism and hospitality but also including other disciplines too, were searched for relevant articles in medical-health-wellness tourism using a set of selected keywords. The ISI Web of Science and Scopus were chosen for this purpose as a result of their international recognition and comprehensiveness. Articles included in the list of references of selected articles were also considered valid as part of this search, in line with methodological suggestions for systematic literature searches [37]. Cited articles were also collected from prominent journals, including the Southern Medical Journal, Journal of Travel Medicine, BMC Public Health, Annals of Tourism Research, Tourism Management, Journal of Travel Research, and Journal of Vacation Marketing. Non-tourism related journals were selected as well including Amfiteatru Economic, Asia Pacific Viewpoint, Public Personal Management, and Revista de Historia Industrial. Adding these references not only delivered a higher number of relevant articles to the database, but it also increased its representativeness.

The second step involved using appropriate, valid and representative search keywords. A total of 986 articles were gathered using the following keywords: medical tourism, health tourism, wellness tourism, and spa tourism. After careful sorting of these publications, using their abstracts and keywords, the number of articles in the database was narrowed down to 802. Of these, 615 were obtained using the keywords medical tourism or wellness tourism, 157 were located by searching for health tourism, and 30 were discovered using spa tourism as the search term. Using the above keywords and restricting the search to 50 years (1970–2020), the first article was found to be published in 1974. As a result, the ensuing analysis of the literature comprises the period from 1974 to 2020. 

### 3.2. Data Analysis

The research tool used for this study was CiteSpace, which is a bibliometric analysis software developed by Professor Chaomei Chen of Drexel University based on the Java framework [38]. This software assists researchers in the analysis of research trends in a specific field of knowledge and presents scientific knowledge structures through visualization. It has been applied to numerous research fields by scholars from many countries. The data processing for this research used the software V.5.7.R2 (64-bit) version.

The data were classified and analyzed to achieve three specific goals. The first and primary goal of this review work was to analyze the content of the chosen articles, including year of publication, authors, journal impact factors, and the institutional affiliations of scholars in this field. The data were then sorted into categories. The order of authorship was not recorded. For multiple-authored articles, each author was given the same level of credit as sole authors. Second, one of the aims of this research was to discover associations in authorships, regions, and affiliations using statistical analysis. Third, the 802 articles were classified into dominant thematic categories applying the approach proposed by Miles and Huberman [39]. Three flows of analytical activities were targeted here: data reduction, data display, and verification of data. In the data reduction activity, the word count technique was adopted. Through content analysis, each article’s title and full-text body were recorded for word counting. The most frequently appearing words were extracted to represent the main topics of the collected articles. The dominant thematic categories to be explored further based on the content analysis and word count were: (1) tourism market: tourist demand and behavior; (2) tourism destinations: development and promotion; and (3) tourism development contexts: policies and impacts.

Finally, in order to refine the set of topic sub-categories, abstracts, first paragraphs, and conclusions were read to make the most appropriate assignments. This approach contributed to the more advanced stages of development of the classification of sub-categories and, consequently, the verification of findings.

## 4. Results

This section presents the results of the data analysis carried out in this study and provides further insights on the methodology adopted.

### 4.1. Overview of Articles Published

The 802 articles selected were all published in English and in international peer-reviewed academic journals. Figure 1 displays the timeline distribution of the research on medical-health-wellness tourism and shows a steady growth in publications in this field between 1974 and 2020. This growth in scholarly activity is particularly significant from 2010 onwards. In fact, 74.9% of the articles were published between 2013 and 2020.

### 4.2. Source Journals

Initially, the first stage of this literature search involved identifying academic journals publishing research articles on medical-health-wellness tourism. It was found that 38 articles had been published on this topic in Tourism Management, and 24 articles in Social Science & Medicine. Table 1 shows the top ten tourism journals for publications in this field, with Tourism Management in first place.

Non-tourism journals in fields such as business, economics, and health, also contributed a significant number of publications in this field, as shown in Table 2.

### 4.3. Author Productivity and Authorship Analysis

The second aim was to identify the most prolific scholars in medical-health-wellness tourism research. This was achieved using co-occurrence network analysis of the authors of relevant research articles (Figure 2). Each node in the co-occurrence map shown in Figure 2 represents a given scholar. The larger the node, the more articles the authors published on the topic, with the connections between nodes representing cooperation between authors.

Among the 2381 authors identified, 1820 (76.4%) contributed to only one article, whereas the remaining 561 (23.6%) authored two or more articles. The three most prolific authors were Jeremy Snyder, Valorie Crooks, and Rory Johnston.

### 4.4. Author Regions and Affiliations

Another objective was to illustrate the relationships and networks of authors publishing research on medical-health-wellness tourism. An analysis of countries this research originated from was carried out using the CiteSpace software. Figure 3 shows that scholars publishing in this field were distributed across 61 countries. The largest group of authors originated from the USA (*n* =197). The second and third largest groups corresponded to Canada (*n* = 88) and the UK (*n* = 84), respectively, followed by Australia (*n* = 70) and South Korea (*n* = 65). As shown in Figure 3, authors from the USA and Canada have made the most significant contributions to medical-health-wellness tourism based on the number of journal articles published.

As shown in Figure 4, a significant number of scholars publishing in this field (*n* = 47) were affiliated to Simon Fraser University in Canada. This university was followed by Sejong University in South Korea (*n* = 13), and the London School of Hygiene & Tropical Medicine (*n* = 13) in the UK. The top universities in terms of author frequency were based in Canada, USA, Australia, UK, South Korea, and Hong Kong.

### 4.5. Thematic Analysis of Research

The fourth research objective was to elicit the prevailing research themes using the 802 articles gathered. First, an analysis of keyword frequency was performed to identify the main research interests. High frequency keywords reflect the research ‘hotspots’ in the field. Using CiteSpace’s keyword visualization analysis function, the keyword co-occurrence knowledge map of medical-health-wellness tourism research was drawn to grasp the research ‘hotspots’ (Figure 5).

Then, content analysis performed on the articles gathered for this study identified three main themes, namely: markets (tourist demand and behavior), destinations (development and promotion), and development environments (policies and impacts). An uneven distribution of research themes is highlighted in Figure 6 and Figure 7.

### 4.6. Markets: Demand and Behavior

Previous studies have shown that the growth of medical-health-wellness tourism in developing countries is largely linked to lower costs, shorter patient waiting lists, and better quality of care [40]. Similarly, it is suggested that the inequalities and failures in domestic health care systems often lead to people seeking treatment to travel abroad to obtain it [41]. In general terms, higher costs, long patient waiting lists, the relative affordability of international air travel, favorable exchange rates, and the availability of well-qualified doctors and medical staff in developing countries, all contribute to this situation [42]. 

As the demand for these forms of tourism has risen over time, processes and factors influencing decision-making have attracted growing levels of scholarly enquiry. For example, a political responsibility model was used to develop a decision-making process for individual medical tourists [43]. A sequential decision-making process has been proposed, including considerations of the required treatments, location of treatment, and quality and safety issues attendant to seeking care [44]. Accordingly, it has been found that health information and the current regulatory environment tend to affect the availability of medical care. 

Multiple factors may simultaneously influence decisions related to the destination for care, including culture [45], social norms [46], religious factors [47], and the institutional environment [48]. It is suggested that socioeconomic conditions shape medical travelers’ decision-making and spending behavior relative to treatment, accommodation, and transport choices as well as the length of stay [49]. Perceived value is a key predictor of tourist intentions. More specifically, perceived medical quality, service quality, and enjoyment significantly influence the intention to travel abroad for medical-health-wellness purposes [50]. Further, perceived quality, satisfaction, and trust in the staff and clinics have significant associations affecting intentions to revisit clinics and the destination country [51]. An empirical study was conducted and found that physical convenience in willingness to stay and time and effort savings in perceived price were key factors affecting the decision-making related to medical hotels [52]. In addition, the level of perceived advantages, price perceptions, and willingness to stay were found to differ significantly between first-time patients and those with two or more previous visits. In addition, it was found that community communication was a major factor influencing decision-making. For instance, it is argued that virtual community membership has a strong influence on tourist behaviors and the way information is transmitted [53]. 

Compared to other tourists, the mental activity and behavior of medical-health-wellness travelers are quite different. Medical tourists are less likely to question their need for surgery and tend to be much readier to accept it [54]. The emotion and anxiety conditions of medical tourists differ from others’ experiences of travel and tourism, as well as their giving and receiving of transnational health care [55]. It has been found that language barriers and parenting responsibilities can be significant challenges, while hospital staff and their own families are often major sources of support for medical tourists [56]. Furthermore, there are significant differences among visitors from different countries in terms of choices, discomfort, preferred product items, and attitudes towards medical tourism [57,58]. 

### 4.7. Destinations: Development and Promotion

In response to the demands of medical-health-wellness tourism, destination development and promotion are attracting growing levels of scholarly interest. Scholars from different countries have discussed the market status of Turkey [12,59], the Caribbean [60] and Barbados [61], India [62,63], Canada [64], and Albania [65]. Table 3 outlines the most frequently researched country destinations in this respect.

The advantages and disadvantages of Turkey were examined and indicated needs for improvements [59]. In another research study, three years (2005, 2007, and 2011) of actual and projected operational cost data were evaluated for three countries: USA, India, and Thailand [66]. This study discussed some of the inefficiencies in the U.S. healthcare system, drew attention to informing uninsured or underinsured medical tourists of the benefits and risks, and determined the managerial and cost implications of various surgical procedures in the global healthcare system. 

As regards medical-health-wellness tourism destination development, scholars have explored research from various perspectives. Conceptual frameworks have been developed to include tourism destinations and services in the context of medical and health tourism [59,67]. Advice has been provided from the perspective of public and private hospital doctors [68]. The principles of designing hospital hotels have been proposed, including proper planning, low prices of tourism services, medical education, creating websites on medical tourism, and health tourism policy councils [69]. Above all, scholars have posited that meeting or exceeding tourist expectations and requirements should remain the top priorities as regards the effective development of medical tourism destinations [69,70].

Once a medical-health-wellness tourism destination is developed successfully, marketing and promotion are essential to attract tourists. As part of this process, informing potential patients about procedural options, treatment facilities, tourism opportunities, and travel arrangements are the keys to success [71]. Most tourists rely on the Internet to gather information about destinations, often using mobile devices or personal computers [72], with websites and social media playing a key role in this respect, and specifically with regards to information about destinations’ medical facilities, staff expertise, services, treatments, equipment, and successful cases [73]. For example, apps for medical travel are available to attract tourists and promote medical tourism in Taiwan [74].

Numerous businesses promote medical-health-wellness travel, including medical travel companies, health insurance companies, travel agencies, medical clinics, and hospitals [75]. Among them, medical travel facilitators play a significant role as engagement moderators between prospective patients in one country and medical facilities elsewhere around the world [76]. The services offered on medical tourism facilitator websites vary considerably from one country to another [77]. Although medical travel facilitators operate on a variety of different scales and market their services differently, they all emphasize the consumer experience through advertising quality assurance and logistical support [78]. 

Scholarly research has also considered the factors that need to be taken into consideration in medical-health-wellness tourism promotion. This research has suggested that destinations should identify the specifics in their health tourism resources, attractions, and products, seek collaboration with others, and build a common regional brand [79]. Regional differences should be considered in the process of marketing as medical-health-wellness tourism is a global industry [77]. International advertisers need to understand the important, contemporary, and cultural characteristics of target customers before promotion [80]. Similarly, destinations need to portray safe and advanced treatment facilities to dispel potential patient worries and suspicions. Messages related solely to low cost may detract from and even undermine messages about quality [71]. However, while benefits are highly emphasized online, websites may fail to report any procedural, postoperative, or legal concerns and risks associated with medical tourism [81].

### 4.8. Development Environments: Policies and Impacts

The rise of medical-health-wellness tourism emphasizes the privatization of healthcare, an increasing dependence on technology, and the accelerating globalization of healthcare and tourism [82]. There are challenges and opportunities in the development of these tourism forms. For instance, it has been suggested that medical tourism distorts national health care systems, and raises critical national economic, ethical, and social questions [83]. Along with the development of medical-health-wellness tourism, social-cultural contradictions [84] and economic inequities are widening in terms of access, cost, and quality of healthcare [85]. It is argued that this tourism leads destinations to emphasize tertiary care for foreigners at the expense of basic healthcare for their citizens [86]. Moreover, in some instances, this phenomenon can exacerbate the medical brain drain from the public sector to the private sector [43,87,88], leading to rising private health care and health insurance costs [88]. 

While medical-health-wellness tourism is a potential source of revenue, it also brings a certain level of risk to destinations and tourists [89]. The spread of this type of tourism has been posited as a contributing factor to the spread of infectious diseases and public health crises [90,91]. Medical tourists are at risk of hospital-associated and procedure-related infections as well as diseases endemic to the countries where the service is provided [92]. Similarly, the safety of some treatments offered has also been the subject of growing levels of scrutiny. Contemporary scholarship examining clinical outcomes in medical travel for cosmetic surgery has identified cases in which patients traveled abroad for medical procedures and subsequently returned home with infections and other surgical complications [93]. Stem cell tourism has been criticized on the grounds of consumer fraud, blatant lack of scientific justification, and patient safety [94,95]. During the process of medical tourism, inadequate communication, and information asymmetry in cross-cultural communication may bring medical risks [96]. 

Medical-health-wellness tourism has emerged as a global healthcare phenomenon. Policy guidance is vital for the development of this sector in the future [97]. There are policy implications for the planning and development of medical-health-wellness tourism destinations [98]. Generally, it has been found that the medical-health-wellness tourism sector tends to perform better in countries with a clear policy framework for this activity [99]. Similarly, scholars have argued the need for a clearer policy framework regulating tourism agencies and the information and services they provide [100]. The upsurge of these tourism forms presents new opportunities and challenges for policy makers in the health sector. It has been argued that existing policy processes are mainly based on entrenched ideological positions and more attention should be paid to robust evidence of impact [101]. The UK developed policies focused on ’patient choice’ that allow people who are able and willing to choose to travel further for healthcare [102]. However, more robust policy making is still required to strengthen national health services and facilitate medical-health-wellness tourism sector development in destinations [103,104]. 

## 5. Discussion and Conclusions

### 5.1. Generation Discussion

This study is based on a literature review of 802 articles on medical-health-wellness tourism from 1970 to 2020. Jeremy Snyder was found to be the most prolific author in this field with 45 articles. It has been found that the literature on this topic can be summarized into three themes: markets (tourist demand and behavior), destinations (development and promotion), and development environments (policies and impacts). The scholarly research in this growing field has undergone a shift in emphasis from tourist demand and behavior to the promotion and development of destinations, and, more recently, to policies and impacts. 

To attract more tourists, destinations should explore their potential for medical-health-wellness tourism. Accessibility, procedural options, treatment facilities, travel arrangements, safety guarantees, and government policies remain influential factors. In the development and promotion of this form of tourism, childhood vaccinations, oral health, legal frameworks, evaluation systems, entrance systems, and macro-policy continue to be areas of concern and where further research is required. Above all, meeting or exceeding tourist expectations and requirements is the most important consideration to promote medical-health-wellness tourism. Similarly, appropriate policy guidelines and frameworks are necessary to support this form of tourism. Importantly, medical-health-wellness tourism may result in negative impacts on the healthcare service provision for local residents in poorer countries, with tourists from richer countries benefiting to the detriment of local communities. However, if managed successfully, this form of tourism can also be a force for good in terms of fostering the economic development of countries delivering these services.

The results indicated that the research literature is spread across a range of different disciplines and there is not one single venue for publishing in this field. A better integration of the research and improved understanding of the overlaps among medical, health, and wellness tourism is required.

### 5.2. Future Research Trends

#### 5.2.1. Industrial Perspective

Medical-health-wellness tourism will, over time, integrate fully with other healthcare and wellness services. Similarly, medical challenges such as disease prevention and traditional medicine remain essential directions for the future of health tourism. This form of tourism will also integrate further with industries such as wellness culinary tourism, mindfulness tourism, active tourism (including adventure tourism), and even cosmetic surgery tourism, leading to a vast array of potential research avenues linked to health tourism destinations. These futures will greatly promote the physical and mental health of wellness tourists. This is another emerging direction for future medical-health-wellness tourism research.

#### 5.2.2. Destination Development Perspectives

Medical-health-wellness tourism will become more significant forms of tourism and impact the development of different nations and areas. For example, this tourism will integrate with Chinese traditional culture. Traditional treatments and remedies will become more of an advantage and should be a topic for future medical-health-wellness tourism research, as well as in other countries with unique health cultures, treatments, and procedures. 

Thailand, Malaysia, and other Southeast Asian countries are favored by tourists from developed countries due to lower costs. In the future, these areas need to focus more on tourism product design, health tourism marketing, community participation, and cross-cultural communication. Developed countries such as the USA, Japan, and South Korea, will use advanced technology and medical equipment to take the path to high-end, high value-added tourism development. This will lead to some new research opportunities.

#### 5.2.3. Tourist Perspectives

Compared with other types of tourists, the needs of medical-health-wellness tourists will receive more attention. Based on previous research, the psychology and perceived value of these tourists are the focus of considerable research. In the future, more emphasis will be paid to people and especially to their psychological and physiological needs. Research on demand will become a more popular topic of this tourism research. Second, the current research on medical-health-wellness tourists is concentrated on the study of tourists in the USA and Canada. Future research should be more dispersed and diversified. Tourists from emerging countries such as Eastern Europe, Asia, the Middle East, and Africa will receive more attention.

### 5.3. Limitations

This study, inevitably, has a number of limitations, including the relatively modest amount of articles collected. Only articles written in English were considered. The sample number is rather small to represent the general research trends in medical-health-wellness tourism from 1970 to 2020. Therefore, it is desirable to increase the number of publications and expand the time and language coverage of the research articles to gain more insights.

Although the research scope of medical-health-wellness tourism is vast, it lacks in-depth exploration. Current research is fragmented, lacks continuity and comprehensiveness, and therefore cannot be considered systematic. Also, the legal aspects of the development of this tourism, environmental capacity of medical-health tourism, wellness tourism management, and mechanisms of profit distribution for medical-health-wellness tourism are less frequently mentioned in research articles. Innovation in this field and international cooperation, and talent cultivation are also not sufficiently addressed. The methods used in medical-health-wellness tourism research are often simple. Scholars still use traditional descriptive statistics and related analysis methods. The theoretical foundation of medical-health-wellness tourism is still relatively weak. We are in the primary stage of this tourism research and in the development of related tourism products. People all over the world are eager for healthy lives. Medical-health-wellness tourism is likely to play a more important future role in travel medicine and tourism research. Beyond what has been done already, follow-up research should be focused on interdisciplinarity and based on the integration of industries. More theoretical research is necessary to support the future growth of medical-health-wellness tourism.

## Figures and Tables

**Figure 1 ijerph-18-10875-f001:**
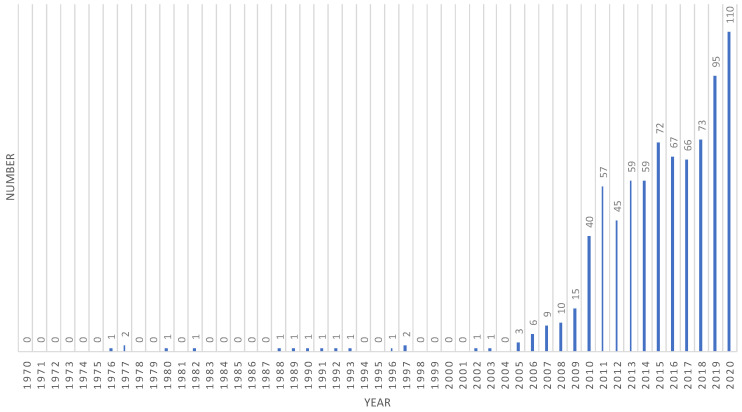
Number of articles by publication year.

**Figure 2 ijerph-18-10875-f002:**
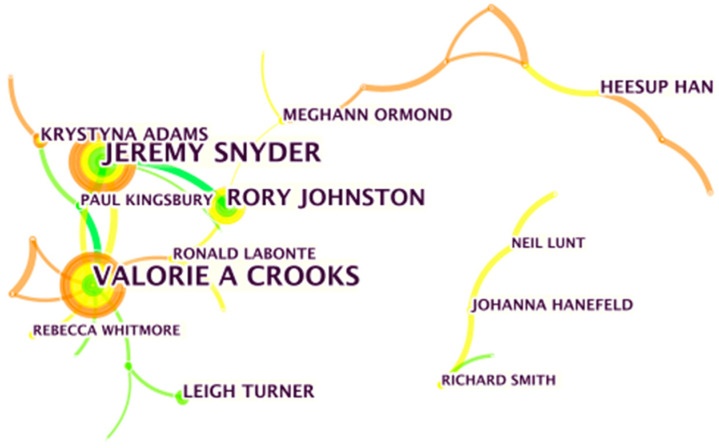
Author article productivity.

**Figure 3 ijerph-18-10875-f003:**
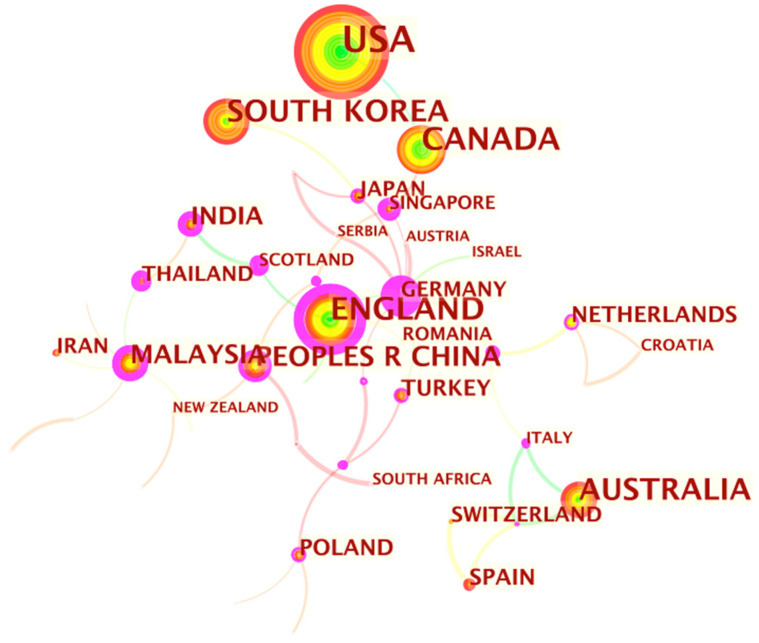
Country of origin of authors in medical-health-wellness tourism.

**Figure 4 ijerph-18-10875-f004:**
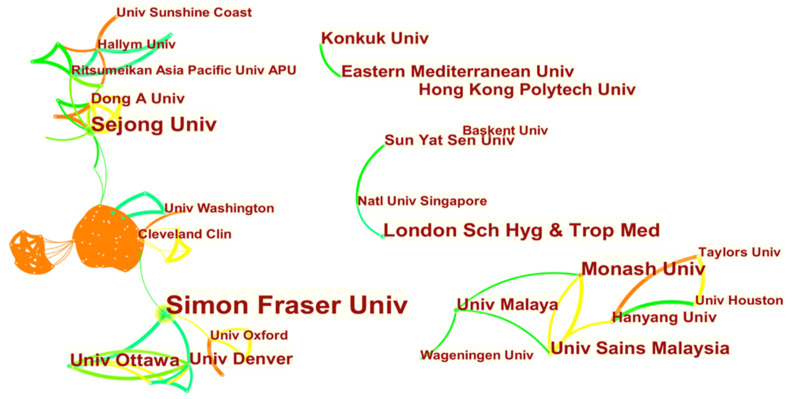
Institutions of authors.

**Figure 5 ijerph-18-10875-f005:**
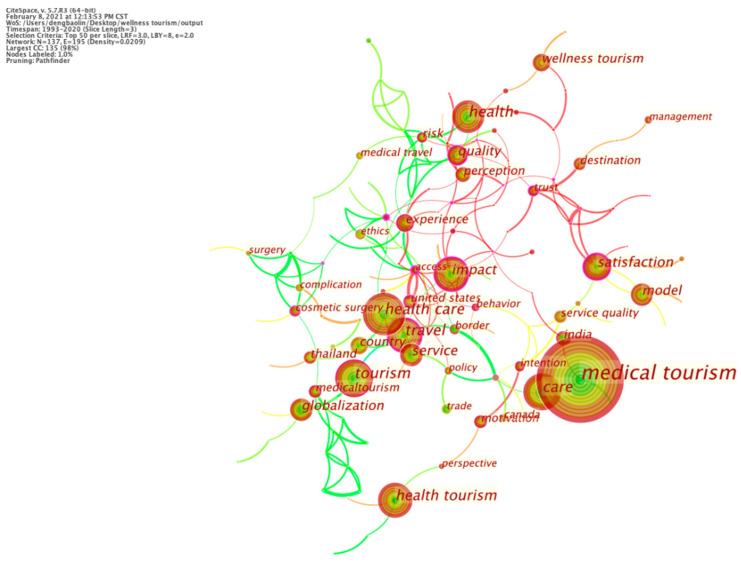
Frequencies of research keywords.

**Figure 6 ijerph-18-10875-f006:**
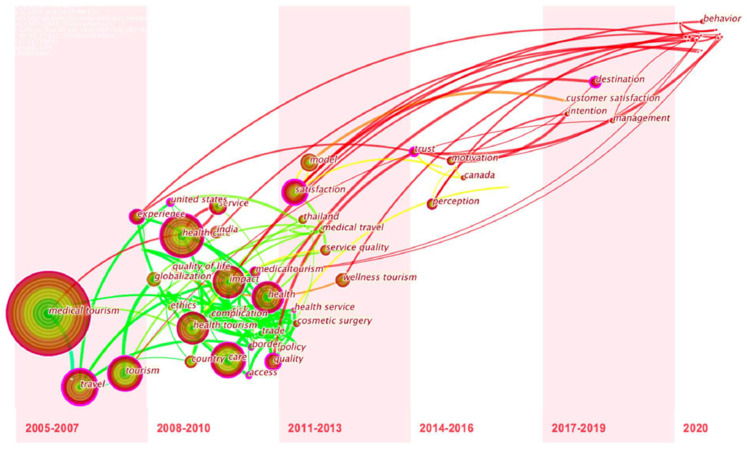
Timeline of research keyword appearance.

**Figure 7 ijerph-18-10875-f007:**
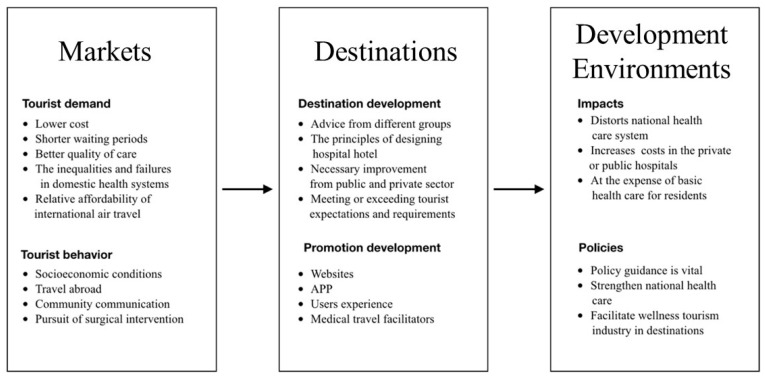
Themes of research articles.

**Table 1 ijerph-18-10875-t001:** Tourism journals publishing articles on medical-health-wellness tourism.

Names of Journals (Top 10)	Number	Percentage
Tourism Management	38	24.67%
Journal of Travel & Tourism Marketing	26	16.88%
Asia Pacific Journal of Tourism Research	22	14.28%
Current Issues in Tourism	18	11.68%
International Journal of Tourism Research	13	8.44%
Annals of Tourism Research	9	5.84%
Journal of Destination Marketing & Management	9	5.84%
Tourism Review	7	4.55%
Journal of Travel Medicine	6	3.90%
Tourism Management Perspectives	6	3.90%
Total	154	100%

**Table 2 ijerph-18-10875-t002:** Non-tourism journals publishing articles on medical-health-wellness tourism.

Names of Journals (Top 10)	Number	Percentage
Social Science & Medicine	24	16.11%
Iranian Journal of Public Health	24	16.11%
Globalization and Health	22	14.77%
Sustainability	22	14.77%
Plastic and Reconstructive Surgery	13	8.72%
BMC Health Services Research	12	8.05%
Canadian Family Physician	9	6.04%
BMJ–British Medical Journal	8	5.37%
Developing World Bioethics	8	5.37%
Journal of Medical Ethics	7	4.70%
Total	149	100%

**Table 3 ijerph-18-10875-t003:** Medical-health-wellness destination frequency in keywords.

Destination	Frequency	Rank
Canada	13	1
India	13	2
Malaysia	9	3
South Korea	9	4
Thailand	8	5
China	7	6
Iran	5	7
Russia	4	8
Singapore	4	9
Taiwan	4	10

## Data Availability

Data are reported in the article.

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
