# Peer review of "Medical, Health and Wellness Tourism Research—A Review of the Literature (1970–2020) and Research Agenda"

_ijerph, 2021, doi:10.3390/ijerph182010875_

Round 1

Reviewer 1 Report

Thank you for giving me the opportunity to review this paper. I have a few comments regarding the reviewed text.

The authors clearly formulated the research objective of the article. However, I have doubts about the data collection process described in the Methods section. The authors gave the reasons why they chose the analyzed journal databases (international recognition and comprehensiveness). Articles collected from some prominent journals were also analyzed. The list of journals includes only a few journals and the authors do not specify (measurable, not subjective) criteria for the selection of journals. This discretionary selection method may affect the results of the analysis. Authors should describe the criteria used for selecting data for analysis to show that these were measurable (objective) criteria, not only the subjective decisions of the authors. 

Author Response

Thanks for your valuable comments.

The authors have provided objective criterial for the selection of data from journals - refer to the revised passages highlighted in yellow on page 3.

Reviewer 2 Report

The paper reviews the literature on medical, health and wellness tourism research using CiteSpace method. The topic discussed in this paper is very interesting and some valuable results and conclusions were drawn. However, I suggest that the work can be improved from the following aspects.

  1. According to the discussion in Section 2, medical tourism, health tourism, and wellness tourism are three different types of tourism. Why are these three topics reviewed together? In addition, when discussing wellness tourism in the fourth paragraph in Section 2, the paper is still talking about medical tourism.
  2. It is suggested that the authors could add some discussion on the results in Section 4.
  3. It is suggested that Section 4.9 be moved to Section 5. Section 4.9 is still about the future research agenda.

Author Response

Thanks for your valuable comments.

The authors have added more discussion on wellness tourism in Section 2. Please refer to the revised passages highlighted in yellow  on page 3 and pages 13-14. The authors added more discussion of the results; however, this was placed in Section 5 under the heading of Discussion and Conclusions (see page 13). Section 4.9 was moved to Section 5 based on your comments.